# Impact of Face-to-Face and Online Mindfulness-Based Public Health Interventions on Health-Related Quality of Life in Older People: A Comparative Study

**DOI:** 10.3390/ijerph22101588

**Published:** 2025-10-20

**Authors:** Denis Juraga, Tomislav Rukavina, Mihaela Marinović Glavić, Darko Roviš, Aleksandar Racz, Lovorka Bilajac, Maša Antonić, Hein Raat, Vanja Vasiljev

**Affiliations:** 1Department of Social Medicine and Epidemiology, Faculty of Medicine, University of Rijeka, 51000 Rijeka, Croatia; tomislav.rukavina@uniri.hr (T.R.); mihaela.marinovic@uniri.hr (M.M.G.); darko.rovis@medri.uniri.hr (D.R.); lovorka.bilajac@uniri.hr (L.B.); vanjav@uniri.hr (V.V.); 2Teaching Institute of Public Health of Primorje-Gorski Kotar County, 51000 Rijeka, Croatia; 3Department of Public Health and Epidemiology, University of Applied Health Sciences, 10000 Zagreb, Croatia; aleksandar.racz@zvu.hr; 4Department of Public Health, Faculty of Health Studies, University of Rijeka, 51000 Rijeka, Croatia; 5Department of Microbiology and Parasitology, Faculty of Medicine, University of Rijeka, 51000 Rijeka, Croatia; masa.antonic@medri.uniri.hr; 6Department of Public Health, Erasmus MC-University Medical Centre Rotterdam, 3015 Rotterdam, The Netherlands; h.raat@erasmusmc.nl

**Keywords:** mindfulness, public health, health-related quality of life, aged

## Abstract

Health-related quality of life (HRQoL) is an important indicator of well-being among older people, especially those living with chronic diseases. Mindfulness-based interventions have shown promise in improving HRQoL. However, in the literature there is a limited number of studies that compare the effectiveness of face-to-face and online mindfulness-based public health interventions. This study aimed to evaluate and compare the effectiveness of face-to-face and online mindfulness-based public health interventions on HRQoL among older people with chronic conditions. A quasi-experimental pre-test–post-test non-randomized study design with non-equivalent groups was conducted among 388 participants aged 65 and older in Rijeka, Croatia. Participants chose to join either a seven-week face-to-face or online mindfulness program or were included in a control group. HRQoL was measured using the SF-12 and EQ-5D-5L questionnaires at baseline and six months post-intervention. Participants in the online intervention showed significant improvements in subjective HRQoL perception regarding physical (*p* < 0.001, η^2^ = 0.066) and mental dimension (*p* = 0.052; η^2^ = 0.010) as well as self-assessed health (EQ-5D-5L = *p* < 0.001, η^2^ = 0.055; EQ-VAS = *p* < 0.001, η^2^ = 0.067) compared to the control group. The face-to-face group also showed improvements, although to a lesser extent. The control group showed no significant change. Both face-to-face and online mindfulness-based interventions may be associated with improvements in HRQoL among older people with chronic conditions, with the online approach demonstrating slightly greater effects. These findings support the use of online approach in community-based public health interventions targeting older populations.

## 1. Introduction

According to the definition provided by the World Health Organization, quality of life (QoL) is “an individual’s perception of their position in life in the context of the culture and value systems in which they live and in relation to their goals, expectations, norms and concerns.” QoL is a multidimensional concept that encompasses not only physical health and well-being, but also a broader range of aspects, including mental health, independence, social relationships, personal beliefs and connectedness to the environment [1]. These domains reflect a person’s subjective perception of quality of life in a cultural, social and environmental context and do not refer exclusively to “health status”,” “lifestyle”,” “life satisfaction” or “well-being”. As quality of life refers to individual perception, it does not involve the “measurement” of symptoms, diseases, conditions or disabilities through an objective assessment. Instead, it encompasses the perceived impact of diseases and public health interventions on a person’s quality of life. Therefore, quality of life is an assessment of a multidimensional concept that includes an individual’s perception of their health, mental and social status, and other aspects of life [1].

Quality of life is often included in public health research as it takes a holistic approach to individuals, encompassing their interaction with the environment and the community in which they live [2]. Historically, the concept of QoL spans several fields, including philosophy, economics and psychology. In the mid-20th century, there was a significant shift in the evaluation of economic indicators and wealth towards individual well-being, happiness and satisfaction. This shift was primarily driven by the realization that economic growth and material wealth are not necessarily associated with higher levels of happiness or satisfaction [3]. Traditional economic indicators such as gross domestic product (GDP) and income levels were recognized as insufficient for a comprehensive understanding of individual and societal well-being. Additional aspects of life, such as health, education, personal and political freedoms, social relationships and environmental conditions, were recognized as integral components of overall societal QoL. This broader perception led to the development of various indicators aimed at capturing non-economic aspects of life that determine its quality, such as the Human Development Index (HDI), the World Happiness Report and the Better Life Index of the Organization for Economic Co-operation and Development (OECD) [4,5,6].

Health-related quality of life (HRQoL) is a key component in the field of public health. It is a comprehensive measure of an individual’s perception of their physical and mental health, life satisfaction and the ability to perform daily activities. This measurement is not limited to the individual, but can also encompass entire communities [7]. In the early development of the HRQoL concept, the focus was primarily on the assessment of disease and disability. Over time, domains of mental health and social functioning were also included [8,9]. Importantly, the HRQoL is characterized by a subjective assessment that emphasizes the individual’s perspective and personal perception of health. It provides an assessment of how a particular state of health or level of disability affects overall well-being. At the community level, the HRQoL assessment includes the resources needed to implement activities, policies and strategies, as well as examples of best practices that impact the health and daily functioning of the community. This information can help authorities to broaden their scope of action in collaboration with other stakeholders from the health and social sectors and the private sector [8]. Today, HRQoL is one of the basic measures for assessing unmet needs in the areas of physical and mental health and the outcomes of interventions. Self-assessed health status is also a strong predictor of mortality and morbidity, among many other objective health measures [10]. This measure provides an evidence-based assessment of the impact of health on QoL. Consequently, HRQoL can assist in self-assessment of chronic diseases (e.g., diabetes, cancer, arthritis and hypertension), risk factors for their development (e.g., body mass index, physical inactivity and smoking), estimation of the burden of preventable disease, injury and disability, and monitoring progress towards health-related goals at the community level [8]. The concepts of general QoL and HRQoL are often considered the same, but it is worth noting that they are completely different aspects. As mentioned above, QoL encompasses all areas of a person’s life (position, culture, value system, etc.), while HRQoL considers health-related changes, relevance to populations with chronic diseases and the effectiveness of public health interventions, and is therefore considered the primary outcome in this type of research.

Public health interventions encompass a wide range of activities that can positively influence various aspects of QoL. For example, community-based interventions, such as the organization of physical activity program, can improve physical health by directly improving individuals’ daily functioning (Hedges’ g = 0.93) and reducing physical pain, which has a positive impact on their QoL [11,12]. Interventions targeting mental health, emotional well-being, self-esteem, independence, social relationships and community engagement can also have a significant positive impact on QoL [11,13,14]. The results of certain public health studies using different approaches of mindfulness-based techniques have shown their positive effects on QoL. These include improved physical health outcomes, a reduction in mental disorders (Cohen’s d = 0.632) and stress (Cohen’s d = 0.205) and improved general well-being [15,16,17,18]. In addition to the face-to-face approach in delivering community-based interventions, over the past two decades, digital services for older people have been widely used in public health research, from telehealth and group video conferencing to online adaptations of evidence-based self-management programs. Randomized and longitudinal studies of internet-based chronic disease self-management (and disease-specific variants such as online diabetes self-management) show feasibility and benefits in mid- to late-life populations, including improvements in health status, activation/self-efficacy and selected clinical outcomes [19,20,21]. Recent reviews also conclude that digital health interventions are feasible for adults ≥ 60 years and can improve self-management behavior and activity levels [22]. There are also studies that have assessed face-to-face approaches versus online formats in older chronic disease populations, where online formats can achieve outcomes comparable to face-to-face approaches. A randomized trial directly comparing an internet-based and face-to-face dyspnea self-management program (mean age ≈ 70 years) found similar improvements in dyspnea and self-efficacy over six months, with no advantage for either modality [23]. Similarly, trials of pulmonary rehabilitation and programs for patients with chronic heart failure report that online-supported or home-based telerehabilitation is non-inferior to conventional center-based programs for functional capacity (6 min walk distance) and symptoms [24,25].

One of the possible approaches to enhance QoL among individuals with chronic conditions is mindfulness which is gaining increasing attention within public health research. There is growing evidence of its effectiveness as an intervention for mental and physical health. Mindfulness is based on traditional Buddhist practices and was introduced to Western medicine and psychology in the late 20th century by Dr Jon Kabat-Zinn and has been widely used in a variety of therapeutic interventions [26]. Although there is no clear definition, mindfulness is the state or ability to observe one’s experience openly and nonjudgmentally, whether it is a simple sensory experience, such as tasting a piece of fruit, or a more complex process of dealing with emotions. It is also described as a state of awareness and intention in the present moment, without thinking about the past or the future and without judgment in relation to thoughts, feelings, bodily sensations and the environment. People are open to new experiences and focus all their attention on their own thoughts and feelings. This affects the neuroplasticity of the brain, i.e., the ability of the nervous system to change its own activity in response to internal or external stimuli by reorganizing its structure, function or neuronal connections and improving cognitive functions [27,28]. There is a growing body of research highlighting the benefits of mindfulness practice, including a meta-analysis by Khoury et al. (2015) [29] that aimed to examine the effectiveness of the Mindfulness-Based Stress Reduction (MBSR) program in different populations such as students, health professionals, pregnant women, general population, etc., who did not suffer from specific diseases or health conditions. Research has shown that mindfulness as a core component of the MBSR program reduces symptoms of depression, anxiety and stress and contributes to a significant reduction in overall stress and an improvement in QoL [29]. In addition, this type of intervention has been shown to help improve chronic physical conditions, including cardiovascular and malignant diseases [30]. In older people, mindfulness practice has been associated with improved cognitive function, better mental health and a higher quality of life [31,32]. While there is growing evidence for the effectiveness of mindfulness-based interventions, some studies suggest that face-to-face approaches to delivering mindfulness-based interventions can be as effective as online formats. A multicenter randomized trial found that internet-based Mindfulness-Based Cognitive Therapy (eMBCT) and face-to-face MBCT were similarly effective in reducing psychological stress (Cohen’s d, 0.45 and 0.71, respectively) and improving mental HRQoL (Cohen’s d, 0.59 and 0.67, respectively) compared with usual care for oncology patients [33]. In chronic pain populations, recent randomized trials show that videoconference mindfulness-based interventions produce meaningful improvements in pain interference and biopsychosocial outcomes and a noninferiority acceptance and commitment therapy (ACT) intervention demonstrated that telehealth delivery was noninferior to in-person care, supporting the plausibility of comparable outcomes with online formats in chronic disease [34,35,36].

The aim of the study was to assess the effectiveness of a seven-week mindfulness-based intervention in a community setting for older people with chronic conditions and to assess its impact on HRQoL. This intervention integrated multiple theoretical and practical frameworks, including salutogenesis, the person-centered approach, positive psychology, and the Transtheoretical Model of behavior change. Additionally, it incorporated mindfulness principles, the GROW coaching model (Goal, Reality, Options, and Will), and elements from two evidence-based programs: the Chronic Disease Self-Management Program (CDSMP) and the Mindfulness-Based Living Program [37,38,39]. In contrast to the standard 8-week MBSR curriculum, which focuses on formal mindfulness practices (e.g., body scan, sitting meditation, and gentle yoga), our integrated program incorporated elements of the CDSMP—such as structured action planning, problem-solving and symptom(s) management skills, and self-efficacy building—and used the GROW (Goal, Reality, Options, and Will) coaching framework to individualize goals and translate mindfulness practices into concrete self-management behaviors for participants with chronic conditions as a novel approach in this kind of public health interventions.

## 2. Materials and Methods

The study utilized a quasi-experimental design for the prospective evaluation of interventions, and compared participants who received the intervention with those who did not. The analysis focused on estimating differences in changes over time in the outcomes of interest between three groups of participants, as it is the case in this study starting from the beginning of the intervention and tracking progress over time [40]. This pre-test–post-test non-randomized study design with non-equivalent groups was conducted from 1 October 2019 to 15 August 2022. During this period, participants were divided into two groups: the intervention group, consisting of individuals who participated in a mindfulness-based intervention, and the control group.

### 2.1. Participants

The participants were residents of the city of Rijeka and the surrounding urban area. Group allocation was determined by the geographical location of the participants and their general practitioners (GPs) or community patronage nurses. Patients who met the inclusion criteria were enrolled consecutively as they expressed interest upon arrival at the GPs or community patronage nurses (first come, first served), who informed them about the project, its activities, and the responsibilities associated with participation in the intervention or control group, until study group-specific targets were reached. A power analysis (two-sided α = 0.05, 80% power) targeting a moderate effect (Cohen’s d = 0.5) on the primary endpoint (change in HRQoL variables from baseline to post-intervention) for each active study group versus control indicated 104 completers per group, allowing for 20% attrition. We aimed to enroll approximately 130 participants per arm, which was achieved.

### 2.2. Inclusion/Exclusion Criteria

The inclusion criteria for participants were individuals of both sexes who were 65 years of age or older, diagnosed with a chronic disease such as cardiovascular disease (e.g., heart failure, hypertension) and/or type II diabetes, living in the city of Rijeka or the surrounding urban area and able to participate in the study for a period of six months. Exclusion criteria included persons without permanent residence in Rijeka or the surrounding urban area, persons who were unable to participate in the six-month study (e.g., persons who planned to be away for a longer period of time or who suffered from a terminal illness), persons experiencing homelessness, persons diagnosed with a mental disorder as defined by the Diagnostic and Statistical Manual of Mental Disorders, Fourth Edition (DSM-IV), persons with cognitive impairments and persons with substance use disorders of alcohol or other addictive substances.

### 2.3. Intervention Procedures

The intervention groups took part in a face-to-face or online seven-week workshop program, with one workshop taking place each week. The first session, entitled “Mind and Body Training for Your Own Wellbeing,” introduced participants to mindfulness and focused on its role in improving emotional and social intelligence and fostering empathy and compassion. This workshop also provided insights into stress management, recognizing habitual “autopilot” behaviors in daily life and identifying personal protective and risk factors. The second workshop, “Healthy Habits”, aimed to raise participants’ awareness of their lifestyle choices. The facilitator presented a model describing the stages of habit change and highlighting strategies to overcome resistance and stress during the process. Mindfulness practices (3 min breathing and body scan) were presented and carried out as a foundation for maintaining awareness of thoughts, emotions, body sensations and the environment to support sustainable habit change. The third session, “Healthy Mindset”, explored the concept of fixed and growth mindsets. Participants were encouraged to cultivate a growth mindset that promotes resilience and focuses on personal development and success with the support of the self-compassion and the 5-4-3-2-1 (grounding) mindfulness practice [41]. In the fourth workshop, “Healthy Eating”, participants were introduced to the principles of mindful eating and healthy eating habits tailored to their health needs. They practiced the mindful eating technique and the 5-4-3-2-1 (grounding) mindfulness practice during the session, with a focus on integrating these practices into their daily routine. The fifth workshop, “Healthy Physical Activity”,” emphasized the importance of daily physical activity and its impact on overall health. Participants were also introduced to the concept of SMART goals (Specific, Measurable, Achievable, Relevant and Time-bound). These SMART goals serve as a practical tool to facilitate the development of ideas, help individuals utilize their time and resources effectively and increase the likelihood of achieving their personal life goals [42]. The participants were also taught to practice mindful walking. The sixth workshop, “Healthy Relationships’,” emphasized the importance of healthy relationships as a fundamental component of health and well-being. Participants explored the concept of emotional intelligence, which is defined as the ability to understand, manage and express one’s own emotions while recognizing and responding to the emotions of others. To support this workshop, participants practiced the mindfulness exercise of loving-kindness, silently repeating phrases of warmth and goodwill to cultivate compassion for themselves and others. The seventh and final workshop, “Healthy Living Despite Chronic Disease,” focused on empowering participants to take an active role in managing their health. Through mindfulness practices (3 min breathing and self-compassion break), participants were encouraged to focus their attention on their personal strengths and resources to minimize the impact of chronic diseases on their health and quality of life. Rather than fixating on the symptoms, the workshop promoted self-management strategies that enabled participants to adopt healthier lifestyle habits such as a balanced diet and regular physical activity.

The face-to-face approach to the implementation of the workshops was organized in such a way that the participants were divided into smaller groups of a maximum of 10 people, taking into account the epidemiological measures that had to be respected during the Coronavirus disease 2019 (COVID-19) pandemic. The workshops took place once a week for two hours. The electronic approach, on the other hand, was based on recordings of the workshops, which were made available to participants via the digital platform YouTube. If necessary, contact with the participants was established via telephone calls, text messages and other communication services such as Viber and WhatsApp. The aim of the seven-week workshop program, led by trained facilitators, was to promote behavioral change among participants by raising their awareness of personal habits and lifestyle choices while addressing modifiable risk factors such as obesity, unhealthy diet, physical inactivity, harmful alcohol consumption and tobacco use. The program also aimed to provide participants with new skills to increase their self-efficacy, boost their self-esteem and improve their ability to take control of their health. In addition, the workshops were designed to promote the development of resilience skills and help participants reduce stress, anxiety and depression [43].

In order to assess the medium-term effects and at the same time equalize the burden on the participants, a 6-month follow-up was conducted. A 6-month window is commonly used in community-based interventions that include chronic disease self-management programs and mindfulness workshops with older people and provides a comparable measure of persistence of change in the outcomes assessed [44,45].

During the study, the determinants that could differentially affect enrollment, participation, and benefit were also considered. That included internet/data access, digital literacy, language and health literacy, sensory or mobility limitations, transport/time costs, and affordability. To reduce barriers, sessions/workshops were offered at flexible times; participants received a brief onboarding call with simple step-by-step guides and ad hoc technical support; and all materials followed plain-language and accessibility principles (e.g., large fonts and captioning when available).

### 2.4. Outcome Measures

To assess the effectiveness of the intervention in terms of HRQoL, a part of the SEFAC questionnaire was used. The questionnaire was developed within the SEFAC project [46]. The reliability of the questionnaires was measured using the internal consistency method and expressed as Cronbach’s α coefficient. Acceptable values of internal consistency of both questionnaires (Cronbach α) were considered to be between 0.70 and 0.95 [47]. The part of the SEFAC questionnaire that assessed HRQoL consisted of two validated and broadly used questionnaire: 12-Item Short-Form Health Survey (SF-12) [48] and the EuroQol-5 Dimensions-5 Level Questionnaire (EQ-5D-5L) [49]. The SF-12 questionnaire is used to measure subjective health or HRQoL in two basic dimensions (physical and mental), namely physical functioning, daily physical limitations, pain, general health, vitality, social functioning, daily emotional limitations and mental health. In the assessment, the results are totaled across the two aforementioned dimensions and accordingly the score can vary between 0 and 100, with a higher score indicating a better QoL. The Cronbach α for self-assessed physical health was 0.88, and 0.87 for the mental health dimension. The EuroQol-5 Dimensions-5 Level Questionnaire (EQ-5D-5L) is a standardized questionnaire for subjective health-related quality of life that is often used to calculate the expected life expectancy reduced by the time lived in disease and disability (Quality-Adjusted Life Year; QALY), with the aim of determining the cost-effectiveness of a particular healthcare intervention. The questionnaire consists of 5 questions relating to mobility, personal hygiene, performance of usual daily activities, pain and anxiety. Based on the participants’ answers, 3.125 different health conditions can be identified with the EQ-5D-5L questionnaire. In addition to the 5 items mentioned above, one item of the EQ visual analog scale (EQ-VAS) was added, with which the study participant rated their own state of health on a scale of 0 to 100 on the day they completed the questionnaire. The internal consistency measure (Cronbach α) for the EQ-5D-5L was 0.81. In addition to the outcome measures, general socio-demographic characteristics were collected, including age, gender, presence of chronic diseases and conditions, marital status, household composition, education level, and household income.

### 2.5. Statistical Analysis

Data were collected at two time points: T0 (baseline measurement) and T1 (six months post-intervention). Statistical analyses were conducted using IBM SPSS Statistics version 28.0.0.0 (IBM Corporation, Armonk, NY, USA). The final analysis included all participants who completed both the baseline and follow-up questionnaires and included the two-way analysis of variance (ANOVA) with repeated measures with post hoc Tukey’s Honestly Significant Difference (HSD) procedure to control the familywise Type I error rate across multiple comparisons. To assess baseline differences between groups, categorical dichotomous variables were analyzed using the chi-square test, and ordinal variables were analyzed using the Kruskal–Wallis test. The results were presented in tables and diagrams. The level of statistical significance was set at α ≤ 0.05.

Figure 1 shows the flow of the study, from recruitment of participants to the follow-up evaluation.

### 2.6. Ethical Considerations

This study was conducted in full compliance with all applicable regulations and guidelines to ensure ethical research practices, participant safety and adherence to the principles of good clinical practice. The research adhered to basic ethical and bioethical principles, including personal integrity (autonomy), justice, beneficence and non-maleficence, in accordance with the Nuremberg Code and the latest revision of the Declaration of Helsinki of the World Medical Association. In addition, the Health Act of the Republic of Croatia (Official Gazette 158/08, 71/10, 139/10, 22/11, 84/11, 12/12, 35/12, 70/12, 82/13, 100/18 and 125/19), the Act on the Rights of Patients of the Republic of Croatia (Official Gazette 169/04, 37/08) and Regulation (EU) 2016/679 of the European Parliament and of the Council of 27 April 2016 on the protection of individuals with regard to the processing of personal data and on the free movement of such data (GDPR) have been complied with.

Ethical approval for the study was granted by the Ethics Committee of the University of Rijeka, Faculty of Medicine (class: 003-08/20-01/91; registration number: 2170-24-09-8-20-3) and the Ethics Committee of the Health Centre of Primorje-Gorski Kotar County, Rijeka, Croatia (registration number: 01-47/2-2-21). The study was registered on 30 August 2018 under ISRCTN registration number ISRCTN11248135 on the UK’s Clinical Study Registry, c/o BMC, The Campus, 4 Crinan Street, London, N1 9XW, UK.

The reporting of this pre-test–post-test non-randomized study was guided by the Transparent Reporting of Evaluations with Nonrandomized Designs (TREND) Statement Checklist [50].

## 3. Results

A total of 388 participants took part in this study: 130 in the face-to-face intervention, 131 in the mindfulness-based online intervention and 127 in the control group. Table 1 shows the socio-demographic characteristics and the primary outcome measures for all three study groups.

As shown in Table 1, the average age of all participants was 71.8. The majority of participants were women, 78.1% of the total sample with no significant statistical difference among the three groups (*p* = 0.664). Female participants outnumbered male participants in all three study groups (*p* ≤ 0.001). Among the participants, 232 participants (59.8% of the total sample) stated that they were married or in a common-law partnership. Almost a third (*n* = 106; 27.3%) were widowed, while 50 participants (12.9%) had never been married with a significant statistical difference between the two intervention and the control group (*p* = 0.027). As the majority of respondents were either married, in some kind of partnership or widowed, more than 70% of participants were not living in a single-person household but shared their household with family members (significant statistical difference among the three groups; *p* = 0.002). In terms of educational level, over 55% of respondents had completed secondary education, including three- or four-year programs or adult education programs and a significant statistical difference was obtained between the three study groups (*p* = 0.018). Most participants (*n* = 264) reported a household income of between €346 and €889, followed by those with an income of between €890 and €1287. No significant statistical difference regarding household income was determined (*p* = 0.080).

No significant differences were observed in the reported levels of subjective HRQoL perception (physical dimension) at the baseline measurement (T0) between participants in the face-to-face and online intervention groups (MD = 0.746, SE = 1.169, 95% CI = [−2.065, 3.557], *p* = 1.000). Similarly, no significant differences were found between these two groups at the follow-up measurement (T1) (MD = −1.846, SE = 1.137, 95% CI = [−4.580, 0.888], *p* = 0.316). Furthermore, at T0, no significant differences were identified in subjective HRQoL perception (physical dimension) between participants in the online intervention group and those in the control group (MD = 0.365, SE = 1.176, 95% CI = [−2.462, 3.193], *p* = 1.000) as well as between the face-to-face and control group participants (MD = 1.111, SE = 1.178, 95% CI = [−1.722, 3.944], *p* = 1.000). However, at T1, six months after the intervention, a significant difference emerged between online intervention group and the control group of participants (MD = 4.438, SE = 1.144, 95% CI = [1.687, 7.188], *p* < 0.001). No significant difference emerged between the face-to-face and control group (MD = 2.592, SE = 1.146, 95% CI = [−0.164, 5.347], *p* = 0.073) in T1. This indicates that participants in the online intervention group reported significantly higher levels of subjective HRQoL perception (physical dimension) compared to the control group at the second measurement point.

The two-way analysis of variance (ANOVA) with repeated measures revealed a significant main effect for the change in subjective HRQoL perception (physical dimension) between the baseline (T0) and second measurement (T1) [F (1, 385) = 11.794, *p* < 0.001, η^2^ = 0.030]. This indicates that participants reported significantly higher levels of subjective HRQoL perception (physical dimension) at T1 compared to T0 regardless of group membership. No significant main effect was found for subjective HRQoL perception (physical dimension) based on group affiliation (face-to-face, online, or control group regardless of measurement point) [F (2, 385) = 2.862, *p* = 0.058, η^2^ = 0.015]. This suggests no significant differences in subjective HRQoL perception (physical dimension) between the intervention and control groups when averaged across time points. However, a significant interaction [F (2, 385) = 8.723, *p* < 0.001, η^2^ = 0.043] between the physical dimension of HRQoL and group affiliation was determined. This means the change in physical HRQoL over time significantly varied between the two measurement time points) depending on the group affiliation (face-to-face approach, online approach and control group of participants). The effect size (η^2^ = 0.043) indicates that the effect of the independent variable on the dependent variable, i.e., the percentage of explained variance, is small to medium in size. The results of the two-way repeated measures ANOVA for the subjective HRQoL perception (physical dimension) are shown in Table 2.

Table 3 shows that there are no significant differences in the reported levels of subjective HRQoL perception (physical dimension) between participants who participated in the face-to-face approach to public health interventions and the control group of participants between the first and second assessment time points, whereas there is a significant difference in the reported levels of subjective HRQoL perception (physical dimension) among participants in the online approach to interventions (*p* < 0.001). Accordingly, participants who attended the online workshop approach reported a higher level of subjective HRQoL perception (physical dimension) at the second measurement (T1) than at the first (T0) measurement.

Figure 2 shows the estimated value in terms of subjective HRQoL perception (physical dimension) in all three groups of participants who participated in the study (face-to-face approach, online approach and control group) and their difference between the first (T0) and the second measurement (T1).

The multivariate analysis showed that the effect size on the subjective HRQoL perception (physical dimension) was largest for the online intervention approach (η^2^ = 0.066), followed by the face-to-face approach (η^2^ = 0.005) and smallest for the control group (η^2^ = 0.001).

No significant differences were observed in the reported levels of subjective HRQoL perception (mental dimension) at the baseline measurement (T0) between participants who participated in the face-to-face and the online form of intervention (MD = 1.630, SE = 1.168, 95% CI = [−1.178, 4.438], *p* = 0.491), between the participants in the online intervention and the control group of participants (MD = −1.062, SE = 1.175, 95% CI = [−3.887, 1.762], *p* = 1.000) as well as between face-to-face and control group participants (MD = 0.568, SE = 1.177, 95% CI = [−2.262, 3.397], *p* = 1.000). At the second measurement time point (T1), there were also no significant differences in the reported levels of the subjective HRQoL perception (mental dimension) between participants who had participated in the face-to-face and the online intervention approach (MD = 1.225, SE = 1.030, 95% CI = [−1.252, 3.702], *p* = 0.705), while at the second measurement point (T1) there was a significant difference in the level of subjective HRQoL perception (mental dimension) between the participants in the online public health interventions and the control group of participants (MD = 2.622, SE = 1.036, 95% CI = [0.131, 5.114], *p* = 0.035) and between face-to-face intervention participants and the control group (MD = 3.847, SE = 1.038, 95% CI = [1.351, 6.343], *p* < 0.001), whereby the participants in the face-to-face and online approach reported a significantly higher level of subjective HRQoL perception (mental dimension) compared to the control group of participants at the second measurement point.

The two-way analysis of variance (ANOVA) with repeated measures showed that there is no main effect of change in subjective HRQoL perception (mental dimension) between the first measurement (T0) and the second measurement time point (T1) [F (1, 385) = 0.181, *p* = 0.670, η^2^ = 0.000], which indicates that there are no differences in the level of subjective HRQoL perception (mental dimension) between the two measurement time points among the participants. The main effect of the change in subjective HRQoL perception (mental dimension) depending on the group affiliation (face-to-face approach, online approach and control group of participants) could also not be confirmed [F (2, 385) = 2.779, *p* = 0.063, η^2^ = 0.014], which means that there is no significant difference between the different groups of participants with regard to subjective HRQoL perception (mental dimension). The analysis also showed a significant interaction effect of the change in subjective HRQoL perception (mental dimension) between the first and second measurement time point depending on the type of intervention [F(2, 385) = 6.233, *p* = 0.002, η^2^ = 0.031), which means that there are differences in the effect of time (assessment of subjective HRQoL perception–mental dimension between the two measurement time points) depending on the group affiliation (face-to-face approach, online approach and control group of participants). The effect size (η^2^ = 0.031) indicates that the effect of the independent variable on the dependent variable, i.e., the percentage of the explained variance, is of a lower magnitude. The results of the two-way repeated measures ANOVA for the subjective HRQoL perception (mental dimension) are shown in Table 4.

Table 5 shows that there are significant differences in the reported levels of subjective HRQoL perception (mental dimension) between the first and second measurement time points in the participants of the online intervention (*p* = 0.052) and in the participants of the control group (*p* = 0.009), while there are no significant differences in the participants of the face-to-face intervention (*p* = 0.151). Therefore, there were no significant changes in subjective HRQoL perception (mental dimension) between the first and second measurements for participants in the face-to-face approach, there was a significant increase in subjective HRQoL perception (mental dimension) for participants in the online approach to interventions, and there was a significant decrease in subjective HRQoL perception (mental dimension) for the control group of participants.

Figure 3 shows the estimated value in terms of subjective HRQoL perception (mental dimension) in all three groups of participants (face-to-face approach, online approach and control group) and their difference between the first (T0) and the second measurement (T1).

Regarding the multivariate analysis and the effect size on subjective HRQoL perception (mental dimension), the largest effect was for the control group of participants (η^2^ = 0.017), followed by the online (η^2^ = 0.010) and the face-to-face approach to the interventions (η^2^ = 0.005).

There are no significant differences between the reported levels of the subjective assessment of HRQoL at the first measurement time point (T0) between the participants who took part in the face-to-face and the online interventions (MD = −0.001, SE = 0.021, 95% CI = [−0.052, 0.050], *p* = 1.000). At the second measurement (T1), there were also no significant differences in the reported levels between the two participant groups (MD = −0.028, SE = 0.021, 95% CI = [−0.078, 0.023], *p* = 0.570). Furthermore, at the first measurement time point (T0), there were no significant differences in the subjective assessment of HRQoL between the participants who took part in the online intervention approach and the control group (MD = 0.006, SE = 0.021, 95% CI = [−0.045, 0.058], *p* = 1.000). When measured six months after the workshops, there was a significant difference in the scores reported between the above groups of participants (MD = 0.067, SE = 0.021, 95% CI = [0.016, 0.117], *p* = 0.005). This indicates that the participants in the online approach had a higher subjective assessment of HRQoL at the second measurement time point (T1) than the participants in the control group. On the other hand, no significant differences were observed between face-to-face participants and the control group at baseline (MD = 0.005, SE = 0.021, 95% CI = [−0.037, 0.048], *p* = 0.802) and at follow-up measurement (MD = 0.039, SE = 0.021, 95% CI = [−0.012, 0.090], *p* = 0.199).

The two-way analysis of variance ANOVA with repeated measures showed that there is a main effect of the subjective assessment of HRQoL between the first measurement (T0) and the second measurement time point (T1) [F (1, 385) = 19.648, *p* < 0.001, η^2^ = 0.049], which means that the participants expressed a different subjective assessment of HRQoL between the two measurement time points in such a way that the research participants expressed significantly higher levels of subjective assessment of HRQoL at the second measurement. On the other hand, the main effect of the change in the subjective assessment of HRQoL depending on the group affiliation (face-to-face approach, online approach and control group of participants) was not confirmed [F (2, 385) = 1.862, *p* = 0.157, η^2^ = 0.010], which means that there is no significant difference between the different groups of participants regarding the subjective assessment of HRQoL. The results of the two-way repeated measures ANOVA for the subjective assessment of HRQoL are shown in Table 6.

These results show a significant interaction effect of the change in the subjective assessment of HRQoL between the first and second measurement time point depending on the type of intervention [F (2, 385) = 4.977, *p* = 0.007, η^2^ = 0.025), which means that there are differences in the effect of time (subjective assessment of HRQoL between the two measurement time points) depending on the group affiliation (face-to-face approach, online approach and control group of participants). The effect size (η^2^ = 0.025) indicates that the effect of the independent variable on the dependent variable, i.e., the percentage of explained variance, is of a smaller magnitude.

Table 7 shows that there are significant differences in the subjective assessment of HRQoL between the first and second measurement time points for participants in the face-to-face and online intervention (*p* = 0.006 and *p* < 0.001), while the control group experienced no significant change (*p* = 0.810). This indicates that the participants of the face-to-face and online intervention approach showed a significantly higher level of subjective assessment of HRQoL between the two measurement time points, while in the control group of participants there was no statistically significant difference in the reported levels of subjective assessment of HRQoL between the T0 and T1 measurements.

Figure 4 shows the estimated values in relation to the subjective assessment of HRQoL in all three groups of participants who took part in the study (face-to-face approach, online approach and control group) as well as the difference between the first (T0) and second measurement (T1).

The multivariate analysis showed that the effect size on the subjective assessment of HRQoL was largest for the online approach (η^2^ = 0.055), followed by the face-to-face approach (η2 = 0.019), and smallest for the control group of subjects (η^2^ = 0.000).

At the first measurement point (T0), there were no significant differences in self-assessed health on the day the questionnaire was completed between the participants who took part in the face-to-face and online interventions (MD = −0.760, SE = 2.083, 95% CI = [−5.769, 4.248], *p* = 1.000), between the participants of the online interventions and the control group of participants (MD = −0.044, SE = 2.095, 95% CI = [−5.082, 4.994], *p* = 1.000) and also between face-to-face and control participants (MD = −0.804, SE = 2.099, 95% CI = [−5.852, 4.243], *p* = 1.000). At the second measurement point (T1), there were also no significant differences in self-assessed health between the participants who took part in the face-to-face and online interventions (MD = −2.060, SE = 2.053, 95% CI = [−6.996, 2.875], *p* = 0.949), while at the second measurement (T1) there was a significant difference in self-assessed health between participants in the online public health intervention approach and the control group of participants (MD = 9.538, SE = 2.065, 95% CI = [4.573, 14.503], *p* < 0.001) as well as between the face-to-face intervention participants and the control group (MD = 7.478, SE = 2.069, 95% CI = [2.503, 12.452], *p* = 0.001). Accordingly, participants in the face-to-face and online approaches reported statistically higher scores on the second measurement in relation to their own health assessment compared to the control group of participants.

After conducting a two-way analysis of variance ANOVA with repeated measures, there was a main effect of change in the assessment of one’s own health on the day of filling out the questionnaire between the initial (T0) measurement and the second measurement time point (T1) [F (1, 385) = 19.120, *p* < 0.001, η^2^ = 0.047], which indicates that the participants expressed different levels of assessment of one’s own health between the two measurement time points with the indication of significantly higher values of assessment of one’s own health between the two measurement points. Also, a main effect of change in the assessment of one’s own health depending on group affiliation (face-to-face approach, online approach and control group of participants) was shown [F (2, 385) = 3.467, *p* = 0.032, η^2^ = 0.018], which means that there is a significant difference between different groups of participants regarding the assessment of one’s own health. Additionally, the analysis showed a significant interaction effect of the change in the assessment of one’s own health between the first and second measurement time point depending on the type of intervention [F (2, 385) = 14.743, *p* < 0.001, η^2^ = 0.071), which means that there are differences in the effect of time (assessment of one’s own health between the two measurement time points) depending on the group affiliation (face-to-face approach, online approach and control group of participants). The effect size (η^2^ = 0.071) indicates that the effect of the independent variable on the dependent variable, i.e., the percentage of explained variance, is of medium size. The results of the two-way analysis of variance ANOVA with repeated measures related to the self-assessed health are shown in Table 8.

Table 9 shows that there are significant differences in the assessment of one’s own health between the first and second measurement time points for participants in the face-to-face and online intervention approach (*p* < 0.001), while the control group of participants experienced no significant change (*p* = 0.061). The results show that the participants of the face-to-face and the online approach of conducting the workshops reported significantly higher values in the assessment of their own health in comparison between the first and the second measurement time point, while the control group of participants did not experience any significant changes between T0 and T1.

Figure 5 shows the estimated values in relation to the assessment of one’s own health for all three groups of participants who took part in the study (face-to-face approach, online approach and control group) as well as the difference between the first (T0) and second measurement (T1).

The multivariate analysis showed that the effect size on self-assessed health was largest for the online interventions (η^2^ = 0.067), followed by the face-to-face approach (η^2^ = 0.045) and smallest for the control group of participants (η^2^ = 0.009).

## 4. Discussion

This study provides evidence that both face-to-face and online mindfulness-based interventions are effective in improving HRQoL among older people with chronic conditions. The results are consistent with previous findings showing mindfulness programs can improve physical and mental health dimensions [18,29].

The pattern of improvements in HRQoL aligns with the complementary pathways targeted by this integrated program. Mindfulness is understood to reduce stress through emotion-regulation mechanisms (attention regulation, decentering, acceptance), which are repeatedly identified as mediators of mindfulness-based interventions (MBIs) effects. In parallel, chronic disease self-management components enhance self-efficacy, problem-solving, and action planning, capabilities associated with better functioning and improved HRQoL in chronic conditions [51,52,53].

Comparative evidence indicates that online mindfulness delivery can be as effective as face-to-face formats for chronic disease. As indicated before, a multicenter randomized controlled trial found that group MBCT and therapist-guided online mindfulness-based cognitive therapy (eMBCT) were similarly effective compared to usual care in improving mental HRQoL [33]. Meta-analyses of online mindfulness-based interventions (MBIs), including updates covering the pandemic, show small to moderate benefits for depression, anxiety, stress, and well-being, which is consistent with the structured guidance in this corresponding study program [54,55]. These convergent data support the plausibility of the slightly larger effects observed in our online intervention group.

Given the circumstances, our study data collection overlapped multiple pandemic waves and varying policy stringency, which likely influenced recruitment, participation, and outcomes. Large multi-country population analyses show that HRQoL, particularly the EQ-5D anxiety/depression dimension, especially during periods of stricter public health measures and lockdowns across 2020–2022; this contextualizes the control group deterioration and underscores the need to consider calendar time in interpretation while highlighting the importance of social support as a key factor for improving deteriorated mental health [56]. The WHO likewise reported a ~25% increase in global anxiety and depression prevalence in the first year of the pandemic, reflecting elevated background stress [57]. At the same time, online modalities often achieved lower no-show rates and better continuity than in-person care during COVID-19, which could enhance effective “dose” and outcomes in online groups [58,59].

Both modes of implementation were associated with an improvement in HRQoL from T0 to T1, with the online study group showing slightly larger effects than the face-to-face group. This pattern may in part reflect the flexibility and convenience of online participation, which reduces barriers commonly experienced by older people (e.g., transport and mobility limitations, simpler scheduling, fatigue, weather circumstances) and is associated with higher pandemic severity/risk perception and infection concerns that discouraged in-person activities during part of the study period [60,61]. Online delivery may also have reduced the number of missed sessions (fewer logistical hurdles) and a higher effective “dose”, while masking/distancing requirements in face-to-face groups may have affected communication and group cohesion—both important for group-based programs [59,62,63]. In addition, access to standardized session materials in online format may have supported adherence to practice and behavior change, as participants could pause, rewind, and revisit core practices between classes, which likely reinforced learning and adherence. The comfort and privacy of the home could have lowered social-evaluative stress and make skills practice (e.g., brief mindfulness exercises, action planning) feel safer and more relevant to everyday contexts [64]. Overall, the observed improvements in self-assessed physical and mental health are consistent with the theoretical underpinnings of the program (salutogenesis, person-centered care and behavior change) which may improve perceived control and self-efficacy in managing chronic conditions. These patterns are consistent with previous reviews and meta-analysis showing that mindfulness-based online interventions (MBIs) have low to moderate benefits in terms of stress and well-being, often comparable to traditional formats, especially when delivered synchronously and with guidance [55,65].

The online approach could also be promoted by local authorities, NGOs and primary health centers in cities and integrated as part of existing community programs in the form of a full online or hybrid model (video conferencing with smartphones) and groups of ~10–15 participants. Key resources that should be included are the training facilitator that has a professional background in delivering mindfulness-based interventions and is certified to deliver this type of approach, digital support in the form of smartphones for the calls on a GDPR-compliant platform, attendance and homework logs, and monitoring of participants in terms of health outcomes before and after the program.

The workshops for the two intervention groups were conducted in different phases of the COVID-19 pandemic and under different restrictions. Increased stress during the pandemic is well documented for different population groups and occupations [66,67,68,69]. Our integrated approach likely acted via complementary pathways: mindfulness improved emotion regulation and reduced stress reactivity, while the self-management component strengthened self-efficacy, action planning and problem-solving mechanisms that may translate into improvements in HRQoL [51,52,70]. As stress and HRQoL are interrelated [60], differences in pandemic conditions at the time of delivery may have influenced the perception of stress and thus both the physical and psychological components of HRQoL in the different study groups. Participants of the online approach to public health interventions could have recorded higher levels of perceived stress in the initial (T0) measurement, which could be partly attributed to the uneven epidemiological situation during the pandemic, which made them have a greater desire and motivation to participate in a program that allows them to reduce stress, increase their quality of life, sense of belonging to the community and social connection. In the control group, the analysis found that there was no significant difference in the physical dimension of subjective HRQoL, subjective assessment of health-related quality of life, and self-assessment of health on the day of completing the questionnaire between the first and second measurements, while there was a significant difference in the mental dimension of subjective health, i.e., a statistically significant deterioration. The explanation for this finding could include the impact of the COVID-19 pandemic itself, since previous research has shown a significant impact of the pandemic on all domains of quality of life as a multidimensional factor [71,72] with an elevated global depression and anxiety burden in 2020 and beyond [73,74,75,76].

### Limitation of the Study

This study has several limitations that should be considered when interpreting the findings. First, the quasi-experimental non-randomized study design with non-equivalent groups limits the ability to establish causal relationships between the interventions and changes in health-related quality of life (HRQoL). Although repeated measures and statistical controls were used, unmeasured confounding variables may still have influenced the outcomes. Second, non-randomized convenient sampling introduces potential selection bias. Individuals’ allocation to a specific study group (first count-first served) who were chosen to participate in mindfulness programs, particularly the online format, may have been more motivated, technologically literate, or health-conscious than those in the control group, possibly inflating observed effects. Third, all outcomes were self-reported, which may be subject to recall and social desirability bias. Objective health indicators, such as physiological measures or medical utilization data, were not collected and could have strengthened the validity of the results. Fourth, the generalizability of the findings may be limited. The study sample consisted primarily of older people from a single urban area in Croatia, with a relatively high proportion of women. Differences in digital access, cultural attitudes toward mindfulness, and health system contexts may limit applicability to other populations. Fifthly, no formal cost–benefit or cost-effectiveness analysis was carried out and the statistical analysis did not include the covariance structure assessed at baseline. Future scale-up should include a prospective economic evaluation (e.g., cost–utility using EQ-5D-5L–derived QALYs) and a budget-impact analysis from the community provider. Sixth, the online group only had access to recordings (no live interaction). This design may have reduced opportunities for peer support, facilitator feedback and group cohesion, mechanisms relevant to mindfulness and self-management programs while simultaneously allowing for repetition of content, which may increase the ‘dose’ compared to face-to-face sessions; both factors complicate the interpretation of modality effects. Seventh, recruitment and follow-up spanned multiple COVID-19 phases and therefore time-varying confounding cannot be ruled out. Finally, the follow-up period was limited to six months post-intervention. Longer-term impacts on HRQoL, health behaviors, and chronic disease management remain unknown.

Future research should address these limitations by (i) delivering randomized controlled trial (RCT) designs to strengthen causal inference, (ii) including longitudinal follow-up to assess the sustainability of intervention effects over 12 months or longer, (iii) exploring objective health outcomes such as biomarkers, clinical measures, or healthcare utilization, (iv) investigating the cost-effectiveness of face-to-face versus online delivery, (v) examining barriers to participation, especially for older people with limited digital literacy, to optimize reach and equity, and (vi) testing hybrid or blended models that combine online and in-person elements to increase flexibility and engagement.

Such studies can provide stronger evidence for integrating scalable mindfulness-based interventions into routine community health and chronic disease management programs, ultimately supporting healthy aging and improved quality of life for diverse older populations.

## 5. Conclusions

This study demonstrates that mindfulness-based public health interventions, delivered either face-to-face or online, were associated with improvements in health-related quality of life among older people with chronic diseases. The findings highlight the potential of flexible, community-based mindfulness programs as accessible strategies to support healthy aging. Future interventions should further adapt delivery methods to individual capabilities and digital literacy levels to maximize reach and impact.

## Figures and Tables

**Figure 1 ijerph-22-01588-f001:**
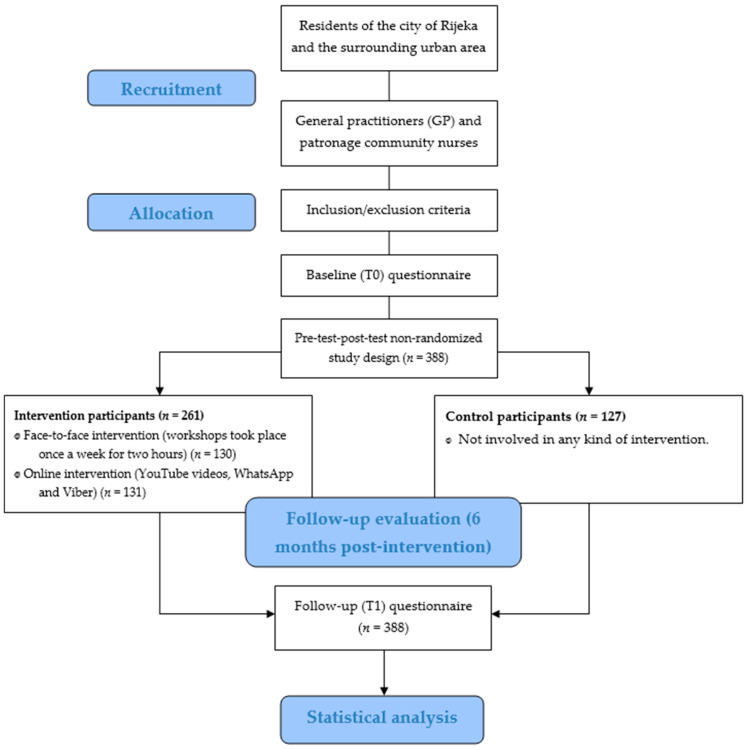
Flow diagram of the study.

**Figure 2 ijerph-22-01588-f002:**
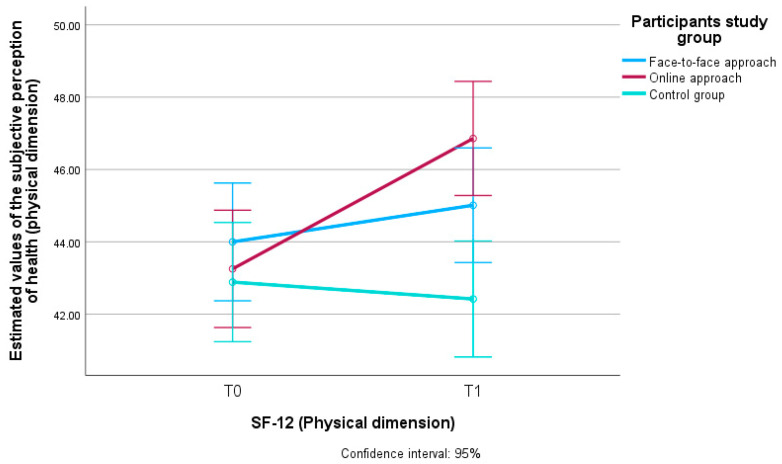
Estimated values of subjective HRQoL perception (physical dimension) in participants of the face-to-face and online intervention approach and the control group at two measurement times (T0 and T1). SF-12 = 12-Item Short-Form Health Survey. T0 = baseline measurement. T1 = follow-up measurement. Confidence interval: 95% = confidence interval and differences with 95% certainty.

**Figure 3 ijerph-22-01588-f003:**
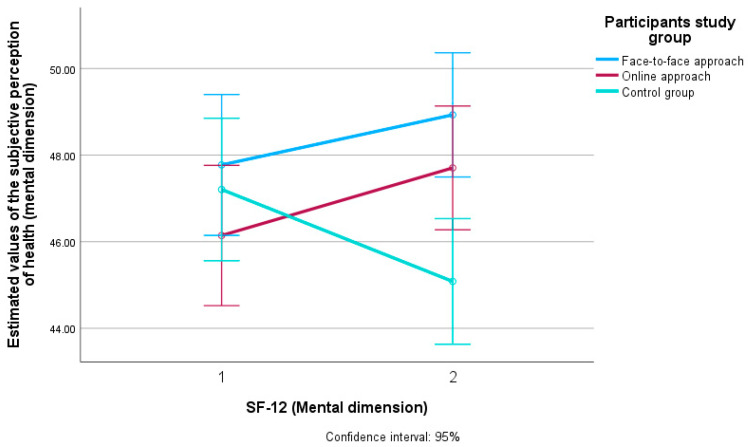
Estimated values of subjective HRQoL perception (mental dimension) in participants of the face-to-face and online intervention approach and the control group at two measurement time points (T0 and T1). SF-12 = 12-Item Short-Form Health Survey. T0 = baseline measurement. T1 = follow-up measurement. Confidence interval: 95% = confidence interval and differences with 95% certainty.

**Figure 4 ijerph-22-01588-f004:**
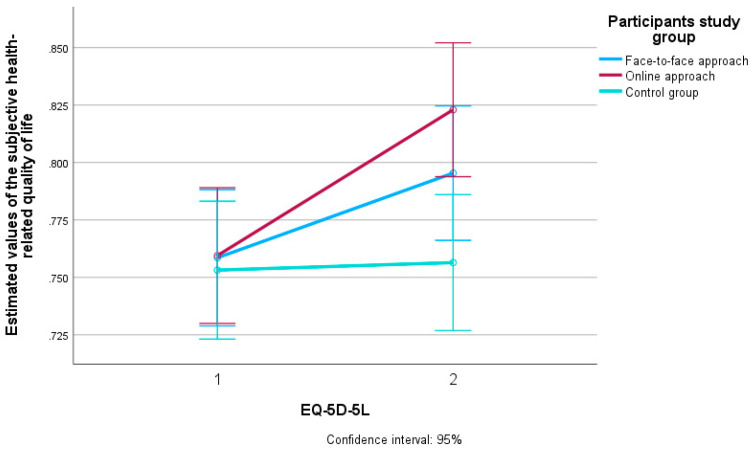
Estimated values of subjective HRQoL in participants of the face-to-face and online intervention method and the control group at two measurement times (T0 and T1). EQ-5D-5L = EuroQol-5 Dimensions-5 Level Questionnaire. T0 = baseline measurement. T1 = follow-up measurement. Confidence interval: 95% = confidence interval and differences with 95% certainty.

**Figure 5 ijerph-22-01588-f005:**
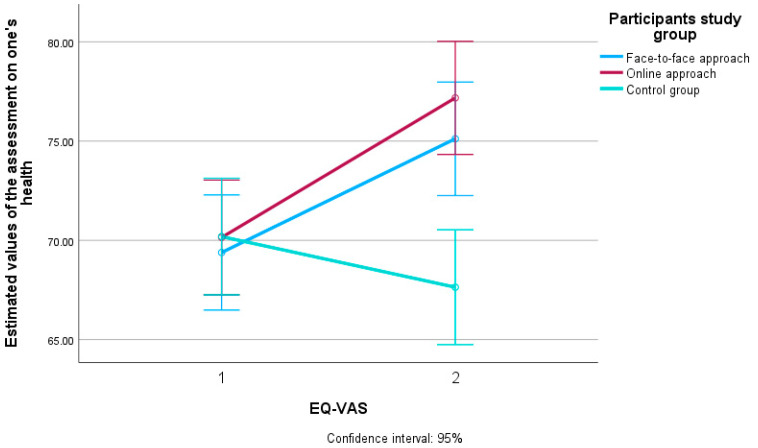
Estimated values of self-assessed health for participants in the face-to-face and online approaches to the interventions and the control group of participants at two measurement time points (T0 and T1). EQ-VAS = EQ Visual Analog Scale. T0 = baseline measurement. T1 = follow-up measurement. Confidence interval: 95% = confidence interval and differences with 95% certainty.

**Table 1 ijerph-22-01588-t001:** Socio-demographic characteristics of all three study groups.

	Face-to-Face Intervention (*n* = 130) *n* (%)	Online Intervention (*n* = 131) *n* (%)	Control Group (*n* = 127) *n* (%)	Total (*n* = 388) *n* (%)	*p*
Age (y) ($\bar{x}$)	71.9	71.5	72.1	71.8	0.664
Sex					
Women Men	116 (89.2)	104 (79.4)	83 (65.4)	303 (78.1)	**<0.001**
	14 (10.8)	27 (20.6)	44 (34.6)	85 (21.9)	
Marital status					
Single Partnership Widow(er)	16 (12.3)	21 (16.0)	13 (10.2)	50 (12.9)	
	65 (50.0) 49 (37.7)	75 (57.3) 35 (26.7)	92 (72.4) 22 (17.3)	232 (59.8) 106 (27.3)	**0.027**
Household composition					
Living alone Living with others	48 (36.9)	42 (32.1)	22 (17.3)	112 (28.9)	
	82 (63.1)	89 (67.9)	105 (82.7)	276 (71.1)	**0.002**
Educational level					
Primary or less Secondary Tertiary or higher	24 (18.5)	4 (3.1)	8 (6.3)	36 (9.3)	
	64 (49.2) 42 (32.3)	75 (57.2) 52 (39.7)	84 (66.1) 35 (27.6)	223 (57.5) 129 (33.2)	**0.018**
Household income					
Decile 1 + 2 (<239–345€) Decile 3 + 4 (346–597€) Decile 5 + 6 (598–889€) Decile 7 + 8 (890–1287€) Decile 9 + 10 (1288–>1606€)	24 (18.5)	13 (9.9)	10 (7.9)	47 (12.1)	0.080
	50 (38.5)	35 (26.7)	55 (43.3)	140 (36.1)	
	26 (20.0)	59 (45.0)	39 (30.7)	124 (32.0)	
	20 (15.4) 10 (7.7)	17 (13.0) 7 (5.3)	19 (15.0) 4 (3.1)	56 (14.4) 21 (5.4)	

*n* = number of participants. % = percentage. y = years. $\bar{x}$ = mean.€ = euro. *p* = level of marginal significance within a statistical hypothesis test, representing the probability of the occurrence of a given event. Bold means the variable is statistically significant.

**Table 2 ijerph-22-01588-t002:** Two-way analysis of variance (ANOVA) with repeated measures related to subjective HRQoL perception (physical dimension).

	Sum of Squares	df	Mean Square	F	*p*	η^2^
SF-12—Physical dimension	371.562	1	371.562	11.794	**<0.001**	0.030
Group: face-to-face, online, control	812.892	2	406.446	2.862	0.058	0.015
SF-12—Physical dimension x Group: face-to-face, online, control	549.607	2	274.803	8.723	**<0.001**	0.043
Error (SF-12—Physical dimension)	12,128.678	385	31.503			
Error (Group: face-to-face, online, control)	54,683.601	385	142.035			

SF-12 = 12-Item Short-Form Health Survey. df = degrees of freedom. F = the ratio of the mean square for the between groups divided by the mean square within groups. *p* = level of marginal significance within a statistical hypothesis test, representing the probability of the occurrence of a given event. η^2^ = partial eta squared. Bold means the variable is statistically significant.

**Table 3 ijerph-22-01588-t003:** Mean values and differences in subjective HRQoL perception (physical dimension) for participants in the face-to-face and online intervention groups and the control group at baseline (T0) and at follow-up (T1).

Participants per Intervention	SF-12—Physical Dimension (T0) $\bar{x}$ (SD)	SF-12—Physical Dimension (T1) $\bar{x}$ (SD)	MD	SE	*p*	95% Confidence Interval for Difference	
						Lower Bound	Upper Bound
Face-to-face (*n* = 130)	44.0 (8.8)	45.0 (8.6)	−1.014	0.696	0.146	−2.382	0.355
Online (*n* = 131)	43.3 (9.8)	46.9 (9.6)	−3.605	0.694	**<0.001**	−4.969	−2.242
Control (*n* = 127)	42.9 (9.7)	42.4 (9.3)	0.467	0.704	0.508	−0.918	1.852

SF-12 = 12-Item Short-Form Health Survey. $\bar{x}$ = mean. SD = standard deviation. *n* = number of participants. T0 = baseline measurement. T1 = follow-up measurement. MD = mean difference between the observed groups. SE = standard error of the mean difference between the observed groups. *p* = level of marginal significance within a statistical hypothesis test, representing the probability of the occurrence of a given event. Bold means the variable is statistically significant.

**Table 4 ijerph-22-01588-t004:** Two-way analysis of variance (ANOVA) with repeated measures related to subjective HRQoL perception (mental dimension).

	Sum of Squares	df	Mean Square	F	*p*	η^2^
SF-12—Mental dimension	7.615	1	7.615	0.181	0.670	0.000
Group: face-to-face, online, control	645.765	2	322.883	2.779	0.063	0.014
SF-12—Mental dimension x Group: face-to-face, online, control	523.541	2	261.771	6.233	**0.002**	0.031
Error (SF-12—Mental dimension)	16,170.276	385	42.001			
Error (Group: face-to-face, online, control)	44,735.846	385	116.197			

SF-12 = 12-Item Short-Form Health Survey. df = degrees of freedom. F = the ratio of the mean square for the between groups divided by the mean square within groups. *p* = level of marginal significance within a statistical hypothesis test, representing the probability of the occurrence of a given event. η^2^ = partial eta squared. Bold means the variable is statistically significant.

**Table 5 ijerph-22-01588-t005:** Mean values and differences in subjective HRQoL perception (mental dimension) for participants in the face-to-face and online intervention groups and the control group at baseline (T0) and at follow-up (T1).

Participants per Intervention	SF-12—Mental Dimension (T0) $\bar{x}$ (SD)	SF-12—Mental Dimension (T1) $\bar{x}$ (SD)	MD	SE	*p*	95% Confidence Interval for Difference	
						Lower Bound	Upper Bound
Face-to-face (*n* = 130)	47.8 (8.8)	48.9 (8.6)	−1.156	0.804	0.151	−2.737	0.424
Online (*n* = 131)	46.1 (10.1)	47.7 (9.6)	−1.561	0.801	**0.052**	−3.136	0.013
Control (*n* = 127)	47.2 (9.4)	45.1 (9.3)	2.123	0.813	**0.009**	0.524	3.722

SF-12 = 12-Item Short-Form Health Survey. $\bar{x}$ = mean. SD = standard deviation. n = number of participants. T0 = baseline measurement. T1 = follow-up measurement. MD = mean difference between the observed groups. SE = standard error of the mean difference between the observed groups. *p* = level of marginal significance within a statistical hypothesis test, representing the probability of the occurrence of a given event. Bold means the variable is statistically significant.

**Table 6 ijerph-22-01588-t006:** Two-way analysis of variance (ANOVA) with repeated measures related to the subjective assessment of HRQoL.

	Sum of Squares	df	Mean Square	F	*p*	η^2^
EQ-5D-5L	0.231	1	0.231	19.648	**<0.001**	0.049
Group: face-to-face, online, control	0.173	2	0.087	1.862	0.157	0.010
EQ-5D-5L x Group: face-to-face, online, control	0.117	2	0.059	4.977	**0.007**	0.025
Error (EQ-5D-5L)	4.533	385	0.012			
Error (Group: face-to-face, online, control)	17.937	385	0.047			

EQ-5D-5L = EuroQol-5 Dimensions-5 Level Questionnaire. df = degrees of freedom. F = the ratio of the mean square for the between groups divided by the mean square within groups. *p* = level of marginal significance within a statistical hypothesis test, representing the probability of the occurrence of a given event. η^2^ = partial eta squared. Bold means the variable is statistically significant.

**Table 7 ijerph-22-01588-t007:** Mean values and differences in subjective assessment of HRQoL for participants in the face-to-face and online intervention groups and the control group at baseline (T0) and at follow-up (T1).

Participants per Intervention	EQ-5D-5L (T0) $\bar{x}$ (SD)	EQ-5D-5L (T1) $\bar{x}$ (SD)	MD	SE	*p*	95% Confidence Interval for Difference	
						Lower Bound	Upper Bound
Face-to-face (*n* = 130)	0.76 (0.15)	0.80 (0.16)	−0.037	0.013	**0.006**	−0.063	−0.010
Online (*n* = 131)	0.76 (0.17)	0.82 (0.16)	−0.063	0.013	**<0.001**	−0.090	−0.037
Control (*n* = 127)	0.75 (0.19)	0.76 (0.19)	−0.003	0.014	0.810	−0.030	0.023

EQ-5D-5L = EuroQol-5 Dimensions-5 Level Questionnaire. $\bar{x}$ = mean. SD = standard deviation. *n* = number of participants. T0 = baseline measurement. T1 = follow-up measurement. MD = mean difference between the observed groups. SE = standard error of the mean difference between the observed groups. *p* = level of marginal significance within a statistical hypothesis test, representing the probability of the occurrence of a given event. Bold means the variable is statistically significant.

**Table 8 ijerph-22-01588-t008:** Two-way analysis of variance (ANOVA) with repeated measures related to the self-assessed health on the day of completing the questionnaire.

	Sum of Squares	df	Mean Square	F	*p*	η^2^
EQ-VAS	2246.703	1	2246.703	19.120	**<0.001**	0.047
Group: face-to-face, online, control	3054.450	2	1527.225	3.467	**0.032**	0.018
EQ-VAS x Group: face-to-face, online, control	3464.734	2	1732.367	14.743	**<0.001**	0.071
Error (EQ-VAS)	45,240.436	385	117.508			
Error (Group: face-to-face, online, control)	169,609.612	385	440.544			

EQ-VAS = EQ Visual Analog Scale. df = degrees of freedom. F = the ratio of the mean square for the between groups divided by the mean square within groups. *p* = level of marginal significance within a statistical hypothesis test, representing the probability of the occurrence of a given event. η2 = partial eta squared. Bold means the variable is statistically significant.

**Table 9 ijerph-22-01588-t009:** Mean values and differences in the expressed assessments of own health on the day of completing the questionnaire between the first (T0) and second (T1) measurement time points depending on group affiliation.

Participants per Intervention	EQ-VAS (T0) $\bar{x}$ (SD)	EQ-VAS (T1) $\bar{x}$ (SD)	MD	SE	*p*	95% Confidence Interval for Difference	
						Lower Bound	Upper Bound
Face-to-face (*n* = 130)	69.4 (15.8)	75.1 (16.9)	−5.731	1.345	**<0.001**	−8.374	−3.087
Online (*n* = 131)	70.1 (17.1)	77.2 (14.9)	−7.031	1.339	**<0.001**	−9.664	−4.397
Control (*n* = 127)	70.2 (17.6)	67.6 (17.9)	2.551	1.360	0.061	−0.123	5.226

EQ-VAS = EQ Visual Analog Scale. $\bar{x}$ = mean. SD = standard deviation. *n* = number of participants. T0 = baseline measurement. T1 = follow-up measurement. MD = mean difference between the observed groups. SE = standard error of the mean difference between the observed groups. *p* = level of marginal significance within a statistical hypothesis test, representing the probability of the occurrence of a given event. Bold means the variable is statistically significant.

## Data Availability

The data presented in this study are available on request from the corresponding author. The data are not publicly available due to privacy and personal data related to the Regulation (EU) 2016/679 of the European Parliament and of the Council of 27 April 2016, on the protection of individuals regarding the processing of personal data and on the free movement of such data.

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
