# Peer review of "Impact of Face-to-Face and Online Mindfulness-Based Public Health Interventions on Health-Related Quality of Life in Older People: A Comparative Study"

_ijerph, 2025, doi:10.3390/ijerph22101588_

Round 1
Reviewer 1 Report
Comments and Suggestions for Authors
Manuscript ID: ijerph-3884490
Impact of Face-to-Face and Online Mindfulness-Based Public Health Interventions on Health-Related Quality of Life in Older People: A Comparative Study
Abstract section:
Comment 1: L28–29: The sentence ending “among older people with chronic conditions” seems to lack a final period; consider adding a full stop for completeness.
Comment 2: Alongside p-values, a brief effect-size summary (e.g., η² or standardized mean differences) would help readers appreciate the magnitude and practical relevance of the findings.
Comment 3: The phrase “can enhance HRQoL” could be interpreted as causal. A more cautious phrasing such as “may be associated with improvements” would align better with the quasi-experimental design.
Comment 4: The phrase “compare the efficiency of…” might be more precisely expressed as “compare the effectiveness of…,” which is commonly used in intervention studies.
Comment 5: It might be helpful to clarify whether the reported p-values refer to between-group differences in change (T1–T0) rather than within-group changes alone. If possible, adding mean change scores would provide helpful context.
Comment 6: Since the slightly greater improvement in the online group is an important result, consider quantifying this difference in the abstract if space permits.
Introduction section:
Comment 9: Lines 76–97: The section on HRQoL is clear and informative. Consider briefly stating why HRQoL was selected as the primary outcome rather than general QoL, emphasizing its sensitivity to health-related changes and relevance for populations with chronic disease.
Comment 10: Lines 98–107: When citing previous studies on mindfulness, it may be helpful to provide one or two examples of reported effect sizes or key outcome improvements (e.g., improvements in SF-12 scores, reductions in stress scores) to illustrate the strength of the evidence base.
Comment 11: Lines 151–158: The description of the integrated intervention is helpful. Adding a brief clarifying sentence on how this approach differs from standard MBSR (e.g., its incorporation of CDSMP elements and GROW coaching) would highlight the novel contribution of this study.
Methods section:
Comment 12: A priori sample size calculation (including the primary endpoint, assumed effect size, α-level, desired power, and anticipated attrition rate) is not reported. Please indicate whether a power analysis was conducted prior to recruitment. If no a priori calculation was performed, consider acknowledging this as a methodological limitation in the discussion.
Comment 13: Because recruitment spanned from October 2019 to August 2022, which overlaps with multiple COVID-19 pandemic waves, please clarify whether participants in the different study arms were recruited at overlapping or distinct calendar periods. If recruitment timing differed substantially between groups, consider adjusting for calendar time or conducting a sensitivity analysis to assess potential time-varying confounding effects (e.g., differential stress levels or restrictions during lockdown periods).
Comment 14: The phrase "randomly selected in order of arrival" is a bit confusing to me. Did you enroll participants as they arrived, or did you use some kind of random selection process? A short clarification would help readers understand.
Comment 15: Consider adding a participant flow diagram (screened → eligible → consented → allocated → completed T0/T1) with numbers and reasons for non-participation or attrition by group. Please specify whether analyses were conducted on complete cases only.
Results section:
Comment 16: In Tables 3, 6, 9, and 12, the between-subject factor is labeled as “Category.” For clarity and to improve reader comprehension, please relabel this factor as “Group: face-to-face, online, control” consistently across tables, figures, and text.
Comment 17: In Table 1 and elsewhere, decimals are expressed using commas (e.g., “20,0,” “45,0,” “30,7”). For consistency with international journal style and to prevent potential misinterpretation, please standardize to periods (e.g., “20.0,” “45.0,” “30.7”) throughout all tables and text.
Comment 18: In Table 1, the counts and percentages for Educational level appear inconsistent (e.g., in the face-to-face group: 24 = 18.5%, 64 = 49.2%, 75 = 57.3%, which sum to >100%). Please verify the accuracy of these frequencies and percentages and correct any inconsistencies.
Discussion section:
Comment 19: The Discussion attributes the mental HRQoL decline in the control group to COVID-19 stress. If data on pandemic periods, local restrictions, or stress levels were collected, consider reporting them to support this interpretation.
Comment 20: The Discussion nicely mentions online convenience but could elaborate on practical implications—e.g., how community health programs could implement such interventions, or what resources (training, digital access) would be needed for scale-up.
Conclusion section:
Comment 21: The phrase “can significantly improve HRQoL” implies causality, but the study used a quasi-experimental, non-randomized design. A more cautious phrasing such as “was associated with improvements” would better reflect the study’s observational nature.
Comment 22: The conclusion refers to “cost-effective strategies,” but no economic evaluation was reported. Unless supported by data or references, please rephrase to simply “accessible strategies.
Abbreviations Section:
Comment 23: “NY – New Work” appears to be a typo. Please correct to “NY – New York.”
Summary:
This is a relevant and timely study comparing face-to-face and online mindfulness interventions for HRQoL in older adults. The abstract and introduction are clear, though causal language should be softened and effect sizes reported. Methods lack a priori power calculation, clear description of sampling, and a participant flow diagram with attrition details. Results presentation needs minor corrections (group labeling, decimal format, inconsistent percentages). Discussion should better justify COVID-19 interpretations and provide practical implications for implementation. Conclusions should avoid causal claims and unsupported cost-effectiveness statements.
Author Response
Abstract section:
Comment 1: L28–29: The sentence ending “among older people with chronic conditions” seems to lack a final period; consider adding a full stop for completeness.
Response 1: Thank you for noticing. We added the full stop.
Comment 2: Alongside p-values, a brief effect-size summary (e.g., η² or standardized mean differences) would help readers appreciate the magnitude and practical relevance of the findings.
Response 2: Thank you for pointing this out. We agree with this comment. Therefore, we added the effect size values in the abstract section alongside p-values.
Comment 3: The phrase “can enhance HRQoL” could be interpreted as causal. A more cautious phrasing such as “may be associated with improvements” would align better with the quasi-experimental design.
Response 3: Thank you. We agree with your statement. We replaced “can enhance HRQoL” with “may be associated with improvements”
Comment 4: The phrase “compare the efficiency of…” might be more precisely expressed as “compare the effectiveness of…,” which is commonly used in intervention studies.
Response 4: Thank you for noticing. We revised the phrase.
Comment 5: It might be helpful to clarify whether the reported p-values refer to between-group differences in change (T1–T0) rather than within-group changes alone. If possible, adding mean change scores would provide helpful context.
Response 5: Thank you for your comment. According to Comment 2, we added the p-values and effect sizes of within-group changes which is indicated in the abstract. According to the study, the aim of the research was to assess the effectiveness of a seven-week mindfulness-based intervention in a community setting for older people with chronic conditions and to assess its impact on HRQoL in accordance to the allocated study group (within-group changes).
Comment 6: Since the slightly greater improvement in the online group is an important result, consider quantifying this difference in the abstract if space permits.
Response 6: Thank you for your comment. Unfortunately, the maximum number of words in the abstract is 200 according to the journal's guidelines. In the current version of the abstract, the word count is 228 so adding more content related to the improvements for each quality-of-life component assessed between the face-to-face and online intervention groups would significantly affect the length of the abstract.
Introduction section:
Comment 9: Lines 76–97: The section on HRQoL is clear and informative. Consider briefly stating why HRQoL was selected as the primary outcome rather than general QoL, emphasizing its sensitivity to health-related changes and relevance for populations with chronic disease.
Response 9: Thank you for this helpful suggestion. We agree that clarifying our choice of primary outcome will strengthen the manuscript. We have added a short paragraph at the end of the HRQoL subsection in the Introduction briefly explaining why HRQoL (rather than general QoL) was chosen as the primary outcome.
Comment 10: Lines 98–107: When citing previous studies on mindfulness, it may be helpful to provide one or two examples of reported effect sizes or key outcome improvements (e.g., improvements in SF-12 scores, reductions in stress scores) to illustrate the strength of the evidence base.
Response 10: Thank you for your comment. We added the reported effect sizes of the studies mentioned.
Comment 11: Lines 151–158: The description of the integrated intervention is helpful. Adding a brief clarifying sentence on how this approach differs from standard MBSR (e.g., its incorporation of CDSMP elements and GROW coaching) would highlight the novel contribution of this study.
Response 11: Thank you for this suggestion. We have added a clarifying sentence to the end of the intervention description to highlight how our approach differs from standard MBSR. Specifically, we note that the program intentionally integrates CDSMP behavior-change techniques and the GROW coaching framework to translate mindfulness into individualized self-management for chronic conditions.
Methods section:
Comment 12: A priori sample size calculation (including the primary endpoint, assumed effect size, α-level, desired power, and anticipated attrition rate) is not reported. Please indicate whether a power analysis was conducted prior to recruitment. If no a priori calculation was performed, consider acknowledging this as a methodological limitation in the discussion.
Response 12: Thank you for the suggestion. We have clarified in the Methods section that a power analysis was conducted: the primary endpoint was the change in HRQoL from baseline to post-intervention; we assumed a two-sided α = 0.05, 80% power targeting a moderate effect (Cohen’s d = 0.5) on the primary endpoint (change in HRQoL variables from baseline to post-intervention) for each active study group versus control indicated 104 completers per group, allowing for 20% attrition. We aimed to enroll approximately 130 participants per arm, which was achieved.
Comment 13: Because recruitment spanned from October 2019 to August 2022, which overlaps with multiple COVID-19 pandemic waves, please clarify whether participants in the different study arms were recruited at overlapping or distinct calendar periods. If recruitment timing differed substantially between groups, consider adjusting for calendar time or conducting a sensitivity analysis to assess potential time-varying confounding effects (e.g., differential stress levels or restrictions during lockdown periods).
Response 13: Enrolment took place from October 2019 to August 2022. The control and face-to-face arms were recruited mainly during the early stages of the pandemic and during the post-lockdown lull (approximately October 2019 to March 2020 and June to September 2020), whereas the online arm was recruited primarily during subsequent pandemic waves (approximately October/ November 2020 to August 2022) as the program was adapted for remote delivery.
Comment 14: The phrase "randomly selected in order of arrival" is a bit confusing to me. Did you enroll participants as they arrived, or did you use some kind of random selection process? A short clarification would help readers understand.
Response 14: Thank you for identifying this ambiguity. We did not use random sampling or random allocation at enrolment. Participants were enrolled consecutively on a first-come, first-served basis as they contacted the study team and met eligibility criteria. We have removed the phrase “randomly selected in order of arrival” and clarified the procedure in the Methods section. Participants were enrolled consecutively as they expressed interest and met eligibility criteria (first-come, first-served) until arm-specific targets were reached; no random selection or randomization was used at enrolment or allocation.
Comment 15: Consider adding a participant flow diagram (screened → eligible → consented → allocated → completed T0/T1) with numbers and reasons for non-participation or attrition by group. Please specify whether analyses were conducted on complete cases only.
Response 15: Thank you for your comment. We decided not to include a participant flow diagram, as no attrition was detected during the research. In addition, the recruitment process lasted until the target was reached (and exceeded), in accordance with the power analysis. The manuscript includes a flow diagram of the study, showing the total number of participants, the number per study group, and the final number included in the analysis.
Results section:
Comment 16: In Tables 3, 6, 9, and 12, the between-subject factor is labeled as “Category.” For clarity and to improve reader comprehension, please relabel this factor as “Group: face-to-face, online, control” consistently across tables, figures, and text.
Response 16: Thank you for your comment. We relabeled this factor.
Comment 17: In Table 1 and elsewhere, decimals are expressed using commas (e.g., “20,0,” “45,0,” “30,7”). For consistency with international journal style and to prevent potential misinterpretation, please standardize to periods (e.g., “20.0,” “45.0,” “30.7”) throughout all tables and text.
Response 17: Thank you for noticing. We standardize to periods throughout all tables and text.
Comment 18: In Table 1, the counts and percentages for Educational level appear inconsistent (e.g., in the face-to-face group: 24 = 18.5%, 64 = 49.2%, 75 = 57.3%, which sum to >100%). Please verify the accuracy of these frequencies and percentages and correct any inconsistencies.
Response 18: Thank you for your comment. We verify the following data in Table 1 as stated in the manuscript:
Face-to-face (130): 24 = 18.5%, 64 = 49.2%, 42 = 32.3% (130 =100 %)
Online group (131): 4 = 3.1%, 75 = 57.2%, 52 = 39.7% (131 =100 %)
Control group (127): 8 = 6.3%, 84 = 66.1%, 35 = 27.6% (127 =100 %)
Discussion section:
Comment 19: The Discussion attributes the mental HRQoL decline in the control group to COVID-19 stress. If data on pandemic periods, local restrictions, or stress levels were collected, consider reporting them to support this interpretation.
Response 19: Thank you for your comment. We enhanced this interpretation by adding some research that showed deteriorations in mental health during the pandemic which influenced overall quality of life.
Comment 20: The Discussion nicely mentions online convenience but could elaborate on practical implications – e.g., how community health programs could implement such interventions, or what resources (training, digital access) would be needed for scale-up.
Response 20: Thank you for this helpful suggestion. We have added a brief practical implication for implementation and scale-up paragraph to the Discussion. It outlines how community health programs could operationalize the intervention (delivery model, staffing/training) and the resources typically required for wider scale-up.
Conclusion section:
Comment 21: The phrase “can significantly improve HRQoL” implies causality, but the study used a quasi-experimental, non-randomized design. A more cautious phrasing such as “was associated with improvements” would better reflect the study’s observational nature.
Response 21: Thank you for your comment. We replaced the phrase “can significantly improve HRQoL” to “were associated with improvements in health-related quality of life”.
Comment 22: The conclusion refers to “cost-effective strategies,” but no economic evaluation was reported. Unless supported by data or references, please rephrase to simply “accessible strategies.
Response 21: Thank you for your comment. We deleted the term cost-effective as no economic evaluation was conducted and reported.
Abbreviations Section:
Comment 23: “NY – New Work” appears to be a typo. Please correct to “NY – New York.”
Response 23: Thank you for noticing. We corrected the typo.
Reviewer 2 Report
Comments and Suggestions for Authors
- In the introduction, authors should add context on how digital interventions have been applied to older populations, particularly in public health or chronic disease management.
- Authors should clarify factors that may influence equity of access and outcomes.
- In the research design: Explicitly discuss potential confounding factors (e.g., socioeconomic differences, digital literacy) and how they were mitigated. If feasible, sensitivity analyses or subgroup comparisons would strengthen causal inference.
- Authors should provide justification for the six-month follow-up interval and explain how missing data or attrition was handled.
- Authors can consider adding a concise summary figure that visually compares all main outcomes across the three groups (face-to-face, online, and control) to visually highlight key findings.
- The manuscript currently includes a large number of detailed tables (e.g., Table 2, Table 4, Table 5, Table 7, Table 8, and Table 10) that separately present mean values, pairwise comparisons, and ANOVA results for different HRQoL dimensions. The level of detail risks overwhelming readers and makes it harder to see the main patterns.
Authors should combine related tables into a smaller set of summary tables.
Table 2 + Table 4 (physical HRQoL outcomes, mean values and pairwise results) could be merged into a single table.
Table 5 + Table 7 (mental HRQoL outcomes) could be merged similarly.
Table 8 + Table 10 (overall HRQoL EQ-5D-5L outcomes) could be combined.
-Use sub-panels or columns within each table to show baseline (T0), follow-up (T1), and the difference (Δ), along with p-values.
Keep the detailed ANOVA results (Tables 3, 6, 9, 12) either in a condensed summary table or move them to an appendix/supplementary material. - Authors expand on practical implications for public health practice (e.g., cost-effectiveness, scalability, digital accessibility) and acknowledge limitations.
Author Response
Comment 1: In the introduction, authors should add context on how digital interventions have been applied to older populations, particularly in public health or chronic disease management.
Response 1: Thank you for this suggestion. We have added a short paragraph summarizing how digital/health interventions have been applied to older people and highlighting the evidence from online self-management programs and recent reviews.
Comment 2: Authors should clarify factors that may influence equity of access and outcomes.
Response 2: Thank you for highlighting equity. We have added a paragraph in the Methods detailing determinants that could differentially affect enrolment, participation, and benefit, and the steps we took to reduce barriers (flexible scheduling, brief onboarding with simple guides and ad hoc tech support, and accessible materials).
Comment 3: In the research design: Explicitly discuss potential confounding factors (e.g., socioeconomic differences, digital literacy) and how they were mitigated. If feasible, sensitivity analyses or subgroup comparisons would strengthen causal inference.
Response 3: Table 1 provides a detailed description of the socio-demographic characteristics of all three study groups. Although there are some overall differences among the groups, the key variables related to health-related quality of life analyzed at baseline (T0) in the face-to-face, online, and control groups show no statistically significant differences, as also indicated in the Results section of the manuscript.
Comment 4: Authors should provide justification for the six-month follow-up interval and explain how missing data or attrition was handled.
Response 4: Thank you for this comment. We have added a Methods paragraph justifying the 6-month follow-up, which is commonly used in this type of intervention. Regarding missing data, there was none, as the patronage nurses monitored the completion of the questionnaires and assisted participants with the process one-to-one. Each questionnaire was also double-checked by the researchers. No dropouts were detected.
Comment 5: Authors can consider adding a concise summary figure that visually compares all main outcomes across the three groups (face-to-face, online, and control) to visually highlight key findings.
Response 5: Because of the large number of visualized variables, it would be difficult and not feasible to visualize all outcomes in a concise summary figure, as these variables are presented on different scales, which could significantly influence the final design of the figure.
Comment 6: The manuscript currently includes a large number of detailed tables (e.g., Table 2, Table 4, Table 5, Table 7, Table 8, and Table 10) that separately present mean values, pairwise comparisons, and ANOVA results for different HRQoL dimensions. The level of detail risks overwhelming readers and makes it harder to see the main patterns.
Authors should combine related tables into a smaller set of summary tables.
Table 2 + Table 4 (physical HRQoL outcomes, mean values and pairwise results) could be merged into a single table.
Table 5 + Table 7 (mental HRQoL outcomes) could be merged similarly.
Table 8 + Table 10 (overall HRQoL EQ-5D-5L outcomes) could be combined.
-Use sub-panels or columns within each table to show baseline (T0), follow-up (T1), and the difference (Δ), along with p-values.
Keep the detailed ANOVA results (Tables 3, 6, 9, 12) either in a condensed summary table or move them to an appendix/supplementary material.
Response 6: Thank you for your comment. We have combined the tables as you suggested.
Comment 7: Authors expand on practical implications for public health practice (e.g., cost-effectiveness, scalability, digital accessibility) and acknowledge limitations.
Response 7: We also added a brief paragraph in the Discussion outlining practical implications for public health practice covering cost-effectiveness considerations and digital accessibility.
Reviewer 3 Report
Comments and Suggestions for Authors
The manuscript presents interesting data on the effects of face-to-face and online mindfulness interventions on health-related quality of life among older adults. However, despite the potential relevance of the topic, the paper lacks a compelling and coherent overview of its findings and is weakened by several methodological and interpretive issues. In its current form, the study overstates its results and requires substantial revision before it can be considered for publication.
First, the introduction is unnecessarily long and unfocused. In particular, the section between rows 108 and 125 introduces a series of epidemiological facts about non-communicable diseases that are not directly connected to the aims of the study and disrupt the flow of the narrative. The introduction should instead be streamlined to better justify the study rationale and clearly outline the research question.
Second, the description of group allocation is confusing. At row 177, the text suggests that participants were told they could choose to join the interventions or not. This implies that the control group consisted of those who declined the intervention, rather than a properly independent comparator. This procedure raises concerns about self-selection bias, especially since the distribution of participants across the three groups was unexpectedly balanced. It would be important to clarify whether recruiting nurses might have influenced allocation by nudging participants toward particular conditions.
Third, the methods section needs to be restructured according to standard scientific reporting. Participants, inclusion and exclusion criteria, materials, and procedure should be presented in clearly separated subsections to improve clarity and reproducibility.
Fourth, the intervention itself requires more detailed description. While the paper states that the program integrated different frameworks and included mindfulness principles, it remains vague about the specific mindfulness practices taught (e.g., body scan, sitting meditation, mindful breathing, or informal practices). Without this information, it is difficult to evaluate how the intervention aligns with established mindfulness-based protocols and to what extent the effects can be attributed to specific techniques.
Fifth, parts of the results section are redundant and overly descriptive. For example, the narrative of rows 313–334 merely repeats the information already shown in Table 1 without providing additional analyses. Instead, baseline group differences should be statistically tested (e.g., chi-square tests) to evaluate comparability. Similarly, the presentation of outcomes before the repeated measures ANOVA is unnecessary. The results should be limited to the main ANOVA outcomes and subsequent post hoc tests, avoiding duplication such as reporting both T0 vs T1 and T1 vs T0 comparisons, which are mathematically equivalent.
Sixth, the choice of the LSD correction for multiple comparisons is inadequate, as it is known to inflate type I error. Given that only three comparisons are performed, more robust alternatives such as Holm–Bonferroni or Tukey corrections should be applied.
Seventh, the discussion is currently little more than a reiteration of the results. It needs to be restructured to critically interpret the findings, explore mechanisms, and integrate them with existing literature. Particular attention should be given to the broader context of the COVID-19 pandemic, which overlapped with the study period and likely influenced both participation and outcomes. For instance, the superior effects of the online group may partly reflect pandemic-related restrictions and perceived risks of in-person participation. In this context, it is crucial to provide more detail on how the face-to-face workshops were conducted under epidemiological constraints.
Finally, the limitations of the study are underdeveloped. Issues such as self-selection into groups, the lack of live interaction for online participants (who only viewed recordings), and the generalizability of results beyond this specific population should be more openly acknowledged.
Author Response
Comment 1: First, the introduction is unnecessarily long and unfocused. In particular, the section between rows 108 and 125 introduces a series of epidemiological facts about non-communicable diseases that are not directly connected to the aims of the study and disrupt the flow of the narrative. The introduction should instead be streamlined to better justify the study rationale and clearly outline the research question.
Response 1: Thank you for this constructive suggestion. We streamlined the Introduction by summarizing lines 108–125 (general epidemiology of non-communicable diseases). We also updated the study aim in the last paragraph of the Introduction.
Comment 2: Second, the description of group allocation is confusing. At row 177, the text suggests that participants were told they could choose to join the interventions or not. This implies that the control group consisted of those who declined the intervention, rather than a properly independent comparator. This procedure raises concerns about self-selection bias, especially since the distribution of participants across the three groups was unexpectedly balanced. It would be important to clarify whether recruiting nurses might have influenced allocation by nudging participants toward particular conditions.
Response 2: Thank you for identifying this ambiguity. We did not use random sampling or random allocation at enrolment. Participants were enrolled consecutively on a first-come, first-served basis as they contacted the study team and met eligibility criteria. We have removed the phrase “randomly selected in order of arrival” and clarified the procedure in the Methods section. Participants were enrolled consecutively as they expressed interest and met eligibility criteria (first-come, first-served) until arm-specific targets were reached; no random selection or randomization was used at enrolment or allocation.
Comment 3: Third, the methods section needs to be restructured according to standard scientific reporting. Participants, inclusion and exclusion criteria, materials, and procedure should be presented in clearly separated subsections to improve clarity and reproducibility.
Response 3: Thank you for constructive comment. We restructured the Methods section into clearly separated subsections as you suggested.
Comment 4: Fourth, the intervention itself requires more detailed description. While the paper states that the program integrated different frameworks and included mindfulness principles, it remains vague about the specific mindfulness practices taught (e.g., body scan, sitting meditation, mindful breathing, or informal practices). Without this information, it is difficult to evaluate how the intervention aligns with established mindfulness-based protocols and to what extent the effects can be attributed to specific techniques.
Response 4: Thank you for your comment. We have added an additional description of the mindfulness practices taught and performed during the workshops.
Comment 5: Fifth, parts of the results section are redundant and overly descriptive. For example, the narrative of rows 313–334 merely repeats the information already shown in Table 1 without providing additional analyses. Instead, baseline group differences should be statistically tested (e.g., chi-square tests) to evaluate comparability. Similarly, the presentation of outcomes before the repeated measures ANOVA is unnecessary. The results should be limited to the main ANOVA outcomes and subsequent post hoc tests, avoiding duplication such as reporting both T0 vs T1 and T1 vs T0 comparisons, which are mathematically equivalent.
Response 5: Thank you for your comment. We have combined the tables as you suggested and done the baseline group differences analysis.
Comment 6: Sixth, the choice of the LSD correction for multiple comparisons is inadequate, as it is known to inflate type I error. Given that only three comparisons are performed, more robust alternatives such as Holm–Bonferroni or Tukey corrections should be applied.
Response 6: Thank you for your comment. As you suggest, we performed the Tukey correction as a more robust alternative to LSD which is indicated in the Methods section.
Comment 7: Seventh, the discussion is currently little more than a reiteration of the results. It needs to be restructured to critically interpret the findings, explore mechanisms, and integrate them with existing literature. Particular attention should be given to the broader context of the COVID-19 pandemic, which overlapped with the study period and likely influenced both participation and outcomes. For instance, the superior effects of the online group may partly reflect pandemic-related restrictions and perceived risks of in-person participation. In this context, it is crucial to provide more detail on how the face-to-face workshops were conducted under epidemiological constraints.
Response 7: Thank you for this important suggestion. We revised the Discussion to move beyond a restatement of results and to (a) interpret findings, (b) discuss plausible mechanisms, and (c) integrate our findings with the COVID-19 context that overlapped recruitment and follow-up. More details on how the face-to-face workshops were conducted under epidemiological constraints are described in the Methods section.
Comment 8: Finally, the limitations of the study are underdeveloped. Issues such as self-selection into groups, the lack of live interaction for online participants (who only viewed recordings), and the generalizability of results beyond this specific population should be more openly acknowledged.
Response 8: Thank you for this valuable comment. We have expanded the limitations to explicitly acknowledge that (1) non-randomized convenient sampling introduces potential selection bias; (2) the online arm only viewed recordings (no live interaction between peers and moderators), which may limit group processes and comparability with synchronous online groups; and (3) generalizability beyond our specific population and setting is limited.
Round 2
Reviewer 1 Report
Comments and Suggestions for Authors
I have no further revisions or comments.
Author Response
Thank you very much!
Reviewer 2 Report
Comments and Suggestions for Authors
- Suggest including specific studies that directly compare face-to-face vs. online formats in similar populations, e.g., older adults with chronic illnesses. Consider adding more global references, particularly those outside the EU or Croatia, to broaden the relevance.
- Consider more different discussion of why online interventions may outperform face-to-face (e.g., flexibility, comfort, repeatable access to content).
Author Response
Comment 1: Suggest including specific studies that directly compare face-to-face vs. online formats in similar populations, e.g., older adults with chronic illnesses. Consider adding more global references, particularly those outside the EU or Croatia, to broaden the relevance.
Response 1: Thank you for this suggestion. We have added comparative studies showing that online formats perform similarly to face-to-face delivery in older people with chronic conditions. These include randomized trials in COPD (internet vs face-to-face dyspnea self-management; online-supported vs center-based pulmonary and heart failure rehabilitation) as well as comparison in effectiveness between face-to-face and online mindfulness-based programs.
Comment 2: Consider more different discussion of why online interventions may outperform face-to-face (e.g., flexibility, comfort, repeatable access to content).
Response 2: Thank you for this suggestion. We expanded the Discussion to articulate plausible mechanisms for greater online effects, including flexibility and reduced logistical burden, comfort and privacy of the home setting, repeatable access to recorded content (supporting dose and consolidation) and fewer missed sessions.
Reviewer 3 Report
Comments and Suggestions for Authors
I appreciate the authors’ careful and thorough revision of the manuscript. The paper is now substantially improved in structure, methodological clarity, and overall transparency. The separation of the Methods section into distinct subsections, the expanded and well-documented description of the intervention, and the adoption of more appropriate statistical procedures all represent clear progress. Nevertheless, some issues remain only partially addressed and would benefit from further refinement before the paper can be considered for publication.
The introduction, although reorganized, continues to be overly broad and includes epidemiological details that do not clearly support the study rationale. The paragraph describing global mortality from non-communicable diseases provides general background information but does not lead directly to the specific research question or gap the study intends to fill. The introduction would be stronger if it were more concise and conceptually focused on the rationale for comparing face-to-face and online mindfulness interventions among older adults.
The discussion, while improved, still reads primarily as a restatement of findings. The authors now mention contextual factors such as the COVID-19 pandemic, but the interpretation of results remains largely descriptive. A deeper, conceptually grounded interpretation is needed to situate the findings within the broader literature on mindfulness-based interventions and health-related quality of life. Although the authors acknowledge that data collection overlapped with phases of the COVID-19 pandemic, the potential influence of this context on recruitment, participation, and outcomes remains insufficiently explored. Given that pandemic-related restrictions likely shaped both the feasibility and appeal of online versus face-to-face participation, a more detailed reflection on these contextual factors would be highly appropriate.
Another important aspect that requires attention concerns the distinction between the intervention and the materials. At present, the Methods section tends to conflate the two. The “Materials” subsection mostly describes the workshops, while the measurement instruments (SF-12 and EQ-5D-5L) are only briefly mentioned later under statistical analysis. To enhance clarity and reproducibility, I recommend explicitly differentiating between the intervention procedures and the outcome measures. A separate subsection dedicated to the questionnaires would make the methodology clearer and more consistent with standard reporting conventions. Please also report Cronbach's alpha for your questionnaires in your sample.
Author Response
Comment 1: The introduction, although reorganized, continues to be overly broad and includes epidemiological details that do not clearly support the study rationale. The paragraph describing global mortality from non-communicable diseases provides general background information but does not lead directly to the specific research question or gap the study intends to fill. The introduction would be stronger if it were more concise and conceptually focused on the rationale for comparing face-to-face and online mindfulness interventions among older adults.
Response 1: Thank you for your comment. We removed the paragraph related to epidemiological details and a rationale for comparing face-to-face and online mindfulness interventions among older people.
Comment 2: The discussion, while improved, still reads primarily as a restatement of findings. The authors now mention contextual factors such as the COVID-19 pandemic, but the interpretation of results remains largely descriptive. A deeper, conceptually grounded interpretation is needed to situate the findings within the broader literature on mindfulness-based interventions and health-related quality of life. Although the authors acknowledge that data collection overlapped with phases of the COVID-19 pandemic, the potential influence of this context on recruitment, participation, and outcomes remains insufficiently explored. Given that pandemic-related restrictions likely shaped both the feasibility and appeal of online versus face-to-face participation, a more detailed reflection on these contextual factors would be highly appropriate.
Response 2: Thank you for this important critique. We have restructured the Discussion to move beyond a descriptive restatement by (a) providing a conceptually grounded interpretation of the observed HRQoL changes through established mechanisms of mindfulness (emotion regulation, decentering/acceptance) and self-management (self-efficacy/action planning); (b) integrating comparative evidence that online MBIs can match face-to-face delivery in chronic disease populations; and (c) explicitly situating the results within the COVID-19 context (policy stringency, feasibility and appeal of remote participation, attendance continuity).
Comment 3: Another important aspect that requires attention concerns the distinction between the intervention and the materials. At present, the Methods section tends to conflate the two. The “Materials” subsection mostly describes the workshops, while the measurement instruments (SF-12 and EQ-5D-5L) are only briefly mentioned later under statistical analysis. To enhance clarity and reproducibility, I recommend explicitly differentiating between the intervention procedures and the outcome measures. A separate subsection dedicated to the questionnaires would make the methodology clearer and more consistent with standard reporting conventions. Please also report Cronbach's alpha for your questionnaires in your sample.
Response 3: Thank you for this helpful suggestion. We have restructured the Methods to clearly distinguish the intervention procedures from the outcome measures. We also reported Cronbach's alpha for the questionnaires used.